# Effect of sampling frequency on fractal fluctuations during treadmill walking

**Vivien Marmelat***, **Austin Duncan, Shane Meltz**

Department of Biomechanics, University of Nebraska at Omaha, Omaha, Nebraska, United States of America

* vmarmelat@unomaha.edu

## Abstract

The temporal dynamics of stride-to-stride fluctuations in steady-state walking reveal important information about locomotor control and can be quantified using so-called fractal analyses, notably the detrended fluctuation analysis (DFA). Gait dynamics are often collected during treadmill walking using 3-D motion capture to identify gait events from kinematic data. The sampling frequency of motion capture systems may impact the precision of event detection and consequently impact the quantification of stride-to-stride variability. This study aimed i) to determine if collecting multiple walking trials with different sampling frequency affects DFA values of spatiotemporal parameters during treadmill walking, and ii) to determine the reliability of DFA values across downsampled conditions. Seventeen healthy young adults walked on a treadmill while their gait dynamics was captured using different sampling frequency (60, 120 and 240 Hz) in each condition. We also compared data from the highest sampling frequency to downsampled versions of itself. We applied DFA to the following time series: step length, time and speed, and stride length, time and speed. Reliability between experimental conditions and between downsampled conditions were measured with 1) intraclass correlation estimates and their 95% confident intervals, calculated based on a single-measurement, absolute-agreement, two-way mixed-effects model (ICC 3,1), and 2) Bland-Altman bias and limits of agreement. Both analyses revealed a poor reliability of DFA results between conditions using different sampling frequencies, but a relatively good reliability between original and downsampled spatiotemporal variables. Collectively, our results suggest that using sampling frequencies of 120 Hz or 240 Hz provide similar results, but that using 60 Hz may alter DFA values. We recommend that gait kinematics should be collected at around 120 Hz, which provides a compromise between event detection accuracy and processing time.

## Introduction

The temporal organization of stride-to-stride fluctuations during steady-state walking can reveal important information about locomotor control [1–6]. With aging and neurodegenerative diseases, gait variability become more random [7–8], compared to the persistent, fractal-like pattern of fluctuations observed in healthy young adults, where large fluctuations are likely

**Funding:** This work was supported by the Center for Research in Human Movement Variability of the

University of Nebraska at Omaha (NIH P20GM109090, https://www.nih.gov/) to VM. The funder had no role in study design, data collection and analysis, decision to publish, or preparation of the manuscript.

**Competing interests:** The authors have declared that no competing interests exist.

to be followed by larger fluctuations, and vice-versa [4,9]. In healthy adults, the temporal organization of fluctuations may also change under different conditions: during metronomic walking (i.e., stepping in time with an auditory metronome), stride time fluctuations become anti-persistent, i.e., large fluctuations are likely to be followed by smaller fluctuations, and vice-versa [1,9]. Similarly, stride length and stride speed become anti-persistent when healthy young adults step on visual targets or walk on a treadmill, respectively [3,10]. More recent studies also evidenced that stride time fluctuations can become more persistent when gait is paced by visual or auditory cues [9–14].

A dominating method to analyze stride-to-stride fluctuations is the detrended fluctuation analysis (DFA) [14], because it provides more accurate results for 'short' time series (<1000 data points) compared to other techniques such as power spectral analysis or rescaled range analysis [15–17]. DFA partitions a time series (e.g., stride time intervals) of length N into non-overlapping windows and calculates the average root mean square (RMS) at each window size. The average RMS at every window size is then plotted against the corresponding window size on a log-log plot. The slope resulting from the line of best fit produces the scaling exponent α-DFA. In an effort to standardize DFA processing, researchers determined some gait-specific parameters required to produce accurate DFA results. Based on both experimental and artificial time series, it is recommended to consider time series of at least 500 data points [16,18–20]. The recommended range of window sizes is 16 to N/9 stride (or step) intervals [21], although for shorter time series a range of 10 to N/4 may be preferred [18,22]. Recent investigations also recommended to use a modified version of the original DFA algorithm, namely the evenly spaced average DFA, to increase the precision of the estimation of the scaling exponent [22–23].

In the context of locomotion, it is also important to consider the parameters underlying data acquisition and pre-processing before applying DFA. In particular, motion capture systems are typically used to record gait kinematics during treadmill walking, but there is no consensus on the most appropriate sampling frequency to reliability apply DFA [24]. While sampling frequency may not have a significant effect on linear measures of gait (e.g., mean and coefficient of variation), it is more likely to influence DFA, because this technique directly depends on the accuracy and precision of gait event detections. In the context of postural control, Rhea et al. [25] found that downsampling linearly decreased the α-DFA scaling exponent of center of pressure (CoP) displacement and CoP velocity. On the other hand, higher sampling frequencies are more likely to introduce artificial white noise (i.e., to decrease α-DFA toward more randomness) [26], and may increase the processing time for little or no benefits.

The goal of this study was to provide guidelines regarding the best sampling frequency to capture fractal dynamics of gait during treadmill walking. We calculated α-DFA values from spatiotemporal variables in different conditions where motion was captured at different sampling frequencies. We compared the average values between conditions, but also the reliability of α-DFA between conditions, using intraclass correlation (ICC) coefficients. Low ICC between different conditions may be due to low between-trial consistency, independently from the sampling frequency. Therefore, we also compared data from a high sampling frequency condition to downsampled data from the same condition.

In summary, this study addressed the following research questions: does motion capture sampling frequency affect α-DFA of spatiotemporal parameters during treadmill walking? What is the reliability of α-DFA values across downsampled conditions? Our central hypothesis was that lower sampling frequency and downsampling will shift α-DFA values toward 0.5, i.e., more randomness due to lower precision in the estimation of gait events.

## Materials and methods

### Participants

Seventeen young adults (Age 23.9 ± 2.7 years, 7 females) were recruited through convenience sampling to participate in the study. All participants self-reported no cognitive, neurological, muscular, or orthopedic impairments. All participants provided written informed consent according to the procedures approved by the local Institutional Review Board at the University of Nebraska Medical Center.

### Equipment

All participants wore their preferred walking shoes and wore a tight-fitting suit. Participants were affixed with 11 retroreflective markers on the following anatomical landmarks to track their motion while walking on a motorized treadmill (Bertec, Columbus, OH): left and right anterior iliac spines, left and right posterior iliac spines, sacrum, dorsal region of the left and right foot between the great toe and long toe, left and right heels, and left and right lateral malleoli. Marker motion was captured through 8 infrared cameras (Vicon, Centennial, CO) at different sampling frequencies in each condition (cf. below).

### Protocol

Participants completed three 15-minute walking trials at their preferred speed. Prior to the trials, individual preferred speed was determined by gradually increasing and decreasing the treadmill speed. The speed at which participants reported being comfortable walking for 15 minutes was selected as their preferred walking speed. Participants were given two minutes to walk at their preferred speed for familiarization before the experimental trials begins. Each trial was collected at a different sampling frequency—60 Hz, 120 Hz, and 240 Hz—in a randomized order. Experimental conditions are described later in this paper by the sampling frequency number (i.e., conditions 60, 120, 240).

### Data processing

Gait events were automatically identified with a custom Matlab function based on the heel, toe, and the average antero-posterior position of hip markers to find the heel strikes and toe offs [27]. We also downsampled the kinematic data from the 240 condition to 120 Hz and 60 Hz (i.e., further referred as DS120 and DS60 conditions, respectively), using Matlab *downsample* function. In this study, we focused on the following spatiotemporal variables from each of the five conditions (three experimental conditions: 60, 120 and 240; two downsampled conditions: DS60 and DS120): step length, stride length, step time, stride time, step speed and stride speed.

Each time series were reduced to the length of the shortest time series (i.e., 740 intervals) for reliable comparisons across participants and conditions. The first 60 step or stride intervals in each time series were removed to reduce the potential confounding effect of gait initiation. Therefore, further analyses considered only 679 step or stride intervals (Fig 1). We calculated the mean, coefficient of variation (CV) and scaling exponent (α-DFA) from each spatiotemporal variable. The scaling exponent was calculated using the evenly spaced average DFA, which was briefly described in the Introduction. We used a range of window from 10 to N/8, where N is the time series length. We selected 18 points in the diffusion plot for the evenly spaced average DFA [24]. An α-DFA value between 0.5 and 1 indicates persistent fluctuations, whereas 0.5 indicates random fluctuations.

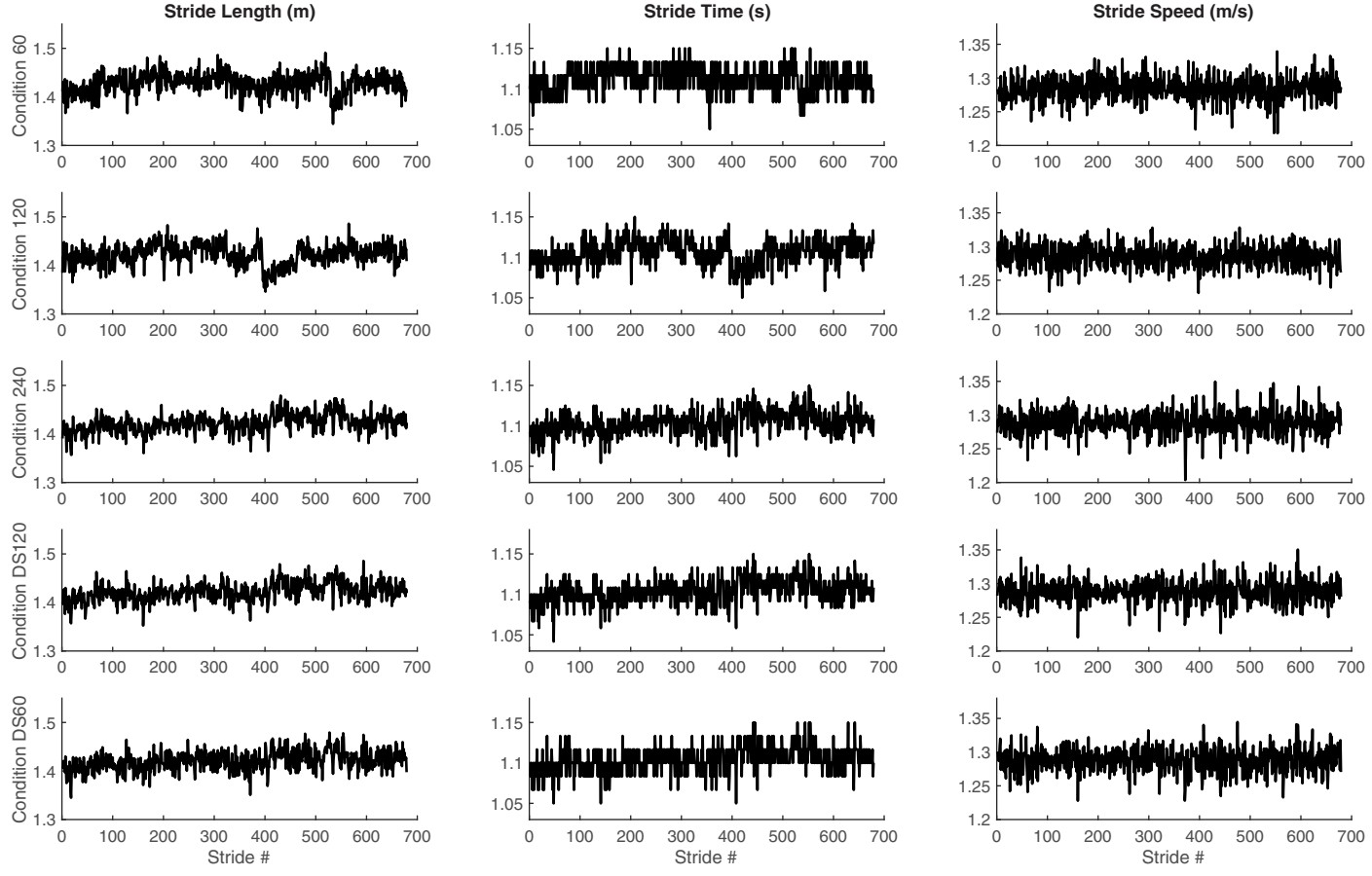

**Fig 1. Time series.** Representative time series from a participant in the three experimental conditions (top three) and the two downsampled conditions (bottom two).

## Statistical analysis

One-way repeated measure ANOVAs were performed 1) between conditions 240, 120, and 60, and 2) between conditions 240, DS120, and DS60 (mean, CV and α-DFA) for each of the six spatiotemporal variables. Post-hoc analysis entailed Tukey's multiple comparison's tests, and the level of statistical significance was set at a $p$-value $< 0.05$.

For each spatiotemporal variable, intraclass correlation (ICC) estimates and their 95% confident intervals were calculated using SPSS statistical package version 23 (SPSS Inc, Chicago, IL) based on a single-measurement, absolute-agreement, two-way mixed-effects model (ICC 3,1) to determine the reliability of mean, CV and α-DFA [28–29]. We compared 1) conditions 240, 120 and 60, and 2) conditions 240, DS120 and DS60. The reliability was graded based on the lower 95% CI values [29], with values less than 0.50 indicating poor reliability, values between 0.50 and 0.75 indicating moderate reliability, values between 0.75 and 0.90 indicating good reliability and values above 0.90 indicating excellent reliability [28].

We also computed Bland-Altman bias and limits of agreement (LoA) on α-DFA for all possible pairs of conditions, for all spatiotemporal variables. Bland-Altman bias and 95% LoA basically assess agreement between two methods [30–31], by studying the mean difference between paired measured (bias), and the agreement interval, within which 95% of the differences of the second method, compared to the first one, fall. The 95% LoA is typically defined

as the bias plus or minus 1.96 the standard deviation of the paired differences for two methods. In this study, we will report LoA, defined as 1.96 the standard deviation of the paired differences for two methods (i.e., to find 95% LoA, simply add or subtract LoA from bias). It is recommended that acceptable limits of agreement should be defined a priori [31]. In the present study, values of LoA equal or below 0.05 for α-DFA were defined as acceptable, based on previous studies using artificial time series to evaluate the proportion of variance due to the computational technique in estimating α-DFA values [16].

## Results

Data from three participants were excluded due to technical difficulties. Data from the remaining 14 participants (Age 23.9 ± 2.8 years, 5 females) were further processed. There was no statistically significant difference between sides for any analyses, so we only report results from the right side in further analyses for the sake of clarity. The spatiotemporal time series from both sides are available in S1 Dataset.

### Effect of sampling frequency

There was no statistically significant difference between any conditions for any measures of any spatiotemporal variables (p>0.05). The ICCs revealed good to excellent reliability for mean of step length and step speed, and excellent reliability for all other spatiotemporal variables (Table 1). Based on the 95% confidence interval, the reliability of CV was poor to good for stride length, step time, stride time and stride speed, and moderate to excellent for step length and step speed. In contrast, for α-DFA the ICC coefficients were poor to good, and the 95% confidence interval revealed poor to moderate reliability for step length, step time, step speed and stride speed, and poor to good reliability for stride length and stride time (Fig 2).

**Table 1. Mean and standard deviation (SD) of time series mean, coefficient of variation (CV) and α-DFA from condition 240, condition 120 and condition 60, and corresponding intraclass correlations and 95% confidence intervals.**

|  |  | Mean (SD) for conditions | | | |
|---|---|---|---|---|---|
|  |  | 240 | 120 | 60 | ICC [95% CI] |
| **Step length** | Mean (m) | 0.63 (0.06) | 0.62 (0.06) | 0.62 (0.07) | 0.915 [0.811–0.969] |
|  | CV (%) | 1.75 (0.64) | 1.81 (0.57) | 2.00 (0.58) | 0.804 [0.585–0.929] |
|  | α-DFA | 0.72 (0.13) | 0.68 (0.09) | 0.67 (0.08) | 0.309 [-0.004–0.652] |
| **Stride length** | Mean (m) | 1.26 (0.11) | 1.26 (0.11) | 1.26 (0.11) | 0.991 [0.979–0.997] |
|  | CV (%) | 1.30 (0.43) | 1.41 (0.51) | 1.45 (0.32) | 0.620 [0.326–0.840] |
|  | α-DFA | 0.77 (0.13) | 0.75 (0.12) | 0.73 (0.11) | 0.536 [0.214–0.797] |
| **Step time** | Mean (s) | 0.54 (0.04) | 0.53 (0.04) | 0.54 (0.04) | 0.992 [0.981–0.997] |
|  | CV (%) | 1.55 (0.47) | 1.67 (0.46) | 1.93 (0.33) | 0.509 [0.175–0.783] |
|  | α-DFA | 0.73 (0.11) | 0.68 (0.09) | 0.65 (0.09) | 0.382 [0.079–0.697] |
| **Stride time** | Mean (s) | 1.07 (0.07) | 1.07 (0.07) | 1.07 (0.07) | 0.993 [0.984–0.998] |
|  | CV (%) | 1.25 (0.54) | 1.28 (0.48) | 1.32 (0.30) | 0.542 [0.222–0.801] |
|  | α-DFA | 0.79 (0.14) | 0.77 (0.10) | 0.77 (0.11) | 0.546 [0.227–0.803] |
| **Step speed** | Mean (m/s) | 1.17 (0.12) | 1.17 (0.12) | 1.15 (0.14) | 0.911 [0.801–0.968] |
|  | CV (%) | 1.74 (0.57) | 1.71 (0.42) | 1.88 (0.39) | 0.861 [0.680–0.950] |
|  | α-DFA | 0.55 (0.06) | 0.56 (0.06) | 0.54 (0.05) | -0.072 [-0.290–0.301] |
| **Stride speed** | Mean (m/s) | 1.18 (0.12) | 1.18 (0.12) | 1.18 (0.12) | 0.998 [0.995–0.999] |
|  | CV (%) | 1.34 (0.84) | 1.17 (0.25) | 1.39 (0.32) | 0.370 [0.040–0.698] |
|  | α-DFA | 0.43 (0.07) | 0.50 (0.20) | 0.53 (0.26) | 0.382 [0.052–0.706] |

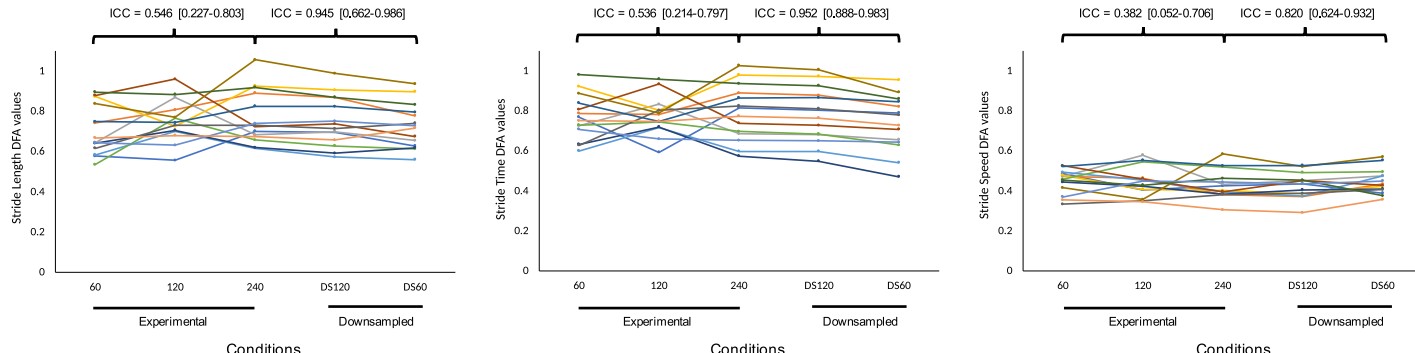

**Fig 2. DFA results.** Individual α-DFA values for stride length (left), stride time (middle) and stride speed (right) in the three experimental conditions and the two downsampled conditions.

Bland-Altman bias and LoA showed similar results (Table 2): for all spatiotemporal variables, there was no consistent bias between conditions 240 and 120, but there was a negative bias between conditions 240 and 60 for step length and stride length α-DFA, indicating lower values in condition 60. Similarly, a small negative bias was observed for most spatiotemporal variables between conditions 120 and 60. In addition, for every pair of conditions, the LoAs ranged from 0.120 to 0.289, well above the acceptable limits of agreement defined at 0.05.

## Effect of downsampling

There was no statistically significant difference between any conditions for any measures of any spatiotemporal variables (p>0.05), except for CV of step time (F(2,39) = 3.917, p = 0.028). The ICCs revealed excellent absolute agreement of means for all spatiotemporal variables (Table 3). For CV, while ICC coefficients were above 0.9 for all spatiotemporal variables, based on the 95% confidence interval the reliability was poor to excellent for step length, moderate to excellent stride length, stride time and step speed, good to excellent for stride speed and excellent for step time. For α-DFA, the 95% confidence interval revealed moderate to excellent

**Table 2. Bland-Altman bias and limits of agreement [LoA], defined as 1.96 standard deviation of the differences, of α-DFA values for each pair of conditions, for all spatiotemporal variables (right side only).** A negative bias indicates that the condition in the top row produced higher α-DFA estimates on average than the condition in the corresponding row.

| Conditions | | 240 | | 120 | | 60 | | DS120 | |
|---|---|---|---|---|---|---|---|---|---|
| Spatiotemporal variable | | Step | Stride | Step | Stride | Step | Stride | Step | Stride |
| 120 | Length | -0.035 [0.268] | -0.016 [0.289] | | | | | | |
| | Time | 0.013 [0.165] | -0.016 [0.276] | | | | | | |
| | Speed | 0.008 [0.182] | 0.013 [0.165] | | | | | | |
| 60 | Length | -0.052 [0.241] | -0.062 [0.175] | -0.017 [0.173] | -0.046 [0.205] | | | | |
| | Time | 0.017 [0.161] | -0.020 [0.158] | -0.030 [0.157] | -0.004 [0.208] | | | | |
| | Speed | -0.011 [0.167] | 0.017 [0.161] | -0.020 [0.132] | 0.004 [0.120] | | | | |
| DS120 | Length | -0.007 [0.035] | -0.019 [0.048] | 0.028 [0.257] | -0.002 [0.273] | 0.045 [0.232] | 0.044 [0.162] | | |
| | Time | -0.004 [0.054] | -0.010 [0.015] | 0.034 [0.256] | 0.006 [0.276] | 0.064 [0.197] | 0.011 [0.156] | | |
| | Speed | -0.005 [0.044] | -0.004 [0.054] | -0.014 [0.173] | -0.017 [0.138] | 0.006 [0.155] | -0.021 [0.133] | | |
| DS60 | Length | -0.040 [0.109] | -0.042 [0.090] | -0.005 [0.253] | -0.025 [0.271] | 0.012 [0.221] | 0.021 [0.162] | -0.033 [0.088] | -0.023 [0.080] |
| | Time | 0.014 [0.084] | -0.052 [0.068] | -0.026 [0.262] | -0.036 [0.280] | 0.004 [0.180] | -0.032 [0.159] | -0.059 [0.088] | -0.042 [0.058] |
| | Speed | -0.011 [0.086] | 0.014 [0.084] | -0.020 [0.151] | 0.001 [0.142] | 0.000 [0.149] | -0.003 [0.143] | -0.006 [0.078] | 0.018 [0.091] |

**Table 3. Mean and standard deviation (SD) of time series mean, coefficient of variation (CV) and α-DFA from condition 240, condition DS120 and condition DS60, and corresponding intraclass correlations and 95% confidence intervals.**

| | | Mean (SD) for conditions | | | |
| --- | --- | --- | --- | --- | --- |
| | | 240 | DS120 | DS60 | ICC [95% CI] |
| **Step length** | Mean (m) | 0.63 (0.06) | 0.63 (0.06) | 0.63 (0.06) | 0.999 [0.998–1] |
| | CV (%) | 1.75 (0.64) | 1.80 (0.61) | 1.98 (0.59) | 0.958 [0.475–0.991] |
| | α-DFA | 0.72 (0.13) | 0.71 (0.11) | 0.68 (0.11) | 0.906 [0.747–0.968] |
| **Stride length** | Mean (m) | 1.26 (0.11) | 1.26 (0.11) | 1.26 (0.11) | 1 [1–1] |
| | CV (%) | 1.30 (0.43) | 1.34 (0.41) | 1.45 (0.39) | 0.959 [0.501–0.991] |
| | α-DFA | 0.77 (0.13) | 0.75 (012) | 0.73 (0.11) | 0.952 [0.888–0.983] |
| **Step time** | Mean (s) | 0.54 (0.04) | 0.54 (0.04) | 0.54 (0.04) | 1 [1–1] |
| | CV (%) | 1.55 (0.47) | 1.65 (0.77) | 1.98 (0.37) | 0.978 [0.946–0.992] |
| | α-DFA | 0.73 (0.11) | 0.72 (0.10) | 0.66 (0.10) | 0.88 [0.736–0.956] |
| **Stride time** | Mean (s) | 1.07 (0.07) | 1.07 (0.07) | 1.07 (0.07) | 1 [1–1] |
| | CV (%) | 1.25 (0.54) | 1.28 (0.53) | 1.41 (0.50) | 0.97 [0.594–0.993] |
| | α-DFA | 0.79 (.14) | 0.78 (0.14) | 0.74 (0.14) | 0.945 [0.662–0.986] |
| **Step speed** | Mean (m/s) | 1.17 (0.12) | 1.17 (0.12) | 1.17 (0.12) | 1 [1–1] |
| | CV (%) | 1.74 (0.57) | 1.79 (0.56) | 1.95 (0.49) | 0.952 [0.532–0.989] |
| | α-DFA | 0.55 (0.06) | 0.54 (0.06) | 0.54 (0.04) | 0.767 [0.537–0.909] |
| **Stride speed** | Mean (m/s) | 1.18 (0.12) | 1.18 (0.12) | 1.18 (0.12) | 1 [1–1] |
| | CV (%) | 1.34 (0.84) | 1.38 (0.83) | 1.58 (0.83) | 0.964 [0.768–0.991] |
| | α-DFA | 0.43 (0.07) | 0.43 (0.06) | 0.45 (0.06) | 0.820 [0.624–0.932] |

reliability for step length, step time, stride time, step speed and stride speed, and good to excellent for stride length (Fig 2). Bland-Altman bias and LoA showed different results for conditions DS120 and DS60 when compared to condition 240. Condition DS120 showed very little bias for every spatiotemporal variable, and the LoAs were all within the acceptable limits of 0.05 (step time and stride speed showed LoA of 0.054, which was still deemed acceptable). In contrast, condition DS60 showed a negative bias, in particular for step length, stride length and stride time. In addition, all the LoA values were above 0.05. Similar results were also found when comparing conditions DS120 to DS60: a negative bias for step length, stride length, step time and stride time, and LoA values above 0.05 for all spatiotemporal variables.

## Discussion

The goal of this study was to determine how motion capture sampling frequency and downsampling procedures affect DFA during treadmill walking. Our four main findings are that i) in general, mean, CV and α-DFA values of all spatiotemporal variables were similar between conditions, as revealed by ANOVAs, whether the data was collected at different sampling frequencies or downsampled, ii) α-DFA values were not reliable between conditions using different sampling frequencies, as revealed by ICCs, iii) α-DFA values were reliable between original and downsampled spatiotemporal variables, in particular between 240 Hz and 120 Hz, as revealed by ICCs and Bland-Altman analyses, and iv) α-DFA from stride intervals were more reliable than α-DFA from step intervals.

Our original hypothesis that lower sampling frequency shift α-DFA values toward more randomness was not supported. We observed a small, non-significant trend toward a reduction in the scaling exponent α-DFA for step length, stride length, step time and stride time, for data originally sampled at 60 Hz or downsampled at 60 Hz. Previous studies have used a range

of sampling frequencies to study gait dynamics during treadmill or overground walking [2,5,21,32–34]. Our results suggest that when the research question focuses on within-group or between-group comparisons, a sampling frequency as low as 60 Hz may be able to capture differences. While the reductions in α-DFA were not significant, 120 Hz may allow for more precise event detection. In addition, walking speed may also play a role: as lower limbs move faster, a greater sampling frequency is needed to capture gait events with the same precision. While this question was beyond the scope of this study and will need to be addressed later, it is an important factor to consider when selecting motion capture sampling frequency. It is also important to note that the number of potential individual values present in a time series depends not only from the sampling frequency, but also from the coefficient of variation (or the range) in that time series. As an illustration, for a stride time series centered around 1-sec with a CV of 5% (i.e., a range of [0.95–1.05]), sampling at 100 Hz would lead to 11 potential values (i.e. 0.95, 0.96, 0.97, etc.). In contrast, a CV of 2% (i.e., a range of [0.98–1.02]) would lead to only 5 potential values and a much more 'squared' signal.

While α-DFA values were not significantly different between conditions, they were not very reliable. Based on the lower 95% confidence intervals, the reliability was graded as poor for all spatiotemporal variables (Table 1). Bland-Altman analyses indicated a similar trend, with limited biases but high limits of agreement, above the a priori defined threshold of 0.05 (Table 2). This is an important finding, as it suggests that collecting data from the same participant using different sampling frequencies would lead to very different scaling exponents in each condition. However, as stressed in the Introduction, a low reliability between conditions may also arise independently from sampling frequencies. While previous studies have shown that α-DFA presented relatively high within-day reliability [33,35–36], it is possible to observe within-subject differences in gait dynamics between conditions. This may arise from different factors such as fluctuations in attention levels, fatigue or habituation to treadmill walking. We anticipated such potential confounding effects, and therefore studied the effect of downsampling (from the highest sampling frequency).

We found that the reliability of α-DFA values graded as moderate and good between original and downsampled spatiotemporal variables (Table 3). Bland-Altman analyses further showed that the data downsampled at 120 Hz provided very similar results as the original data sampled at 240 Hz, with no consistent bias and limits of agreement below the threshold of 0.05. This suggests that using a sampling frequency of 240 Hz does not provide more benefit than 120 Hz to capture the 'true' α-DFA values. In contrast, the Bland-Altman bias was higher between conditions 240 and DS60, and the limits of agreement were above 0.05 for all spatio-temporal variables. This suggests that sampling motion capture at 60 Hz (i.e., as in condition DS60) may lead to less accurate α-DFA values, assuming condition 240 as the gold-standard. These results contrast with our previous finding (comparing different conditions), and suggests that the low reliability observed between conditions sampled at different frequencies originated from within-subject differences more than reflecting a true effect of sampling frequency.

It should also be stressed that α-DFA from stride intervals were more reliable than step intervals. This may be because a single stride interval 'encompasses' two step intervals (i.e., one from each side). Therefore, small corrections occurring at the step level may not be reflected in a more global stride interval.

Our study presents several limitations. First, our final sample size (N = 14) may be relatively small, considering that each participant underwent three conditions [37]. While the results from intraclass correlations and Bland-Altman analyses lead to similar conclusions, the small sample size remains a major limitation of the present study. We also collected only healthy young adults, as in previous methodological studies, because healthy gait patterns are often

used as a reference [21,32–33,35–36]. We cannot exclude the possibility that the results would be different with other populations such as older adults or people with gait disorders. Another limitation of our study is that we only considered three different sampling frequencies. While technically motion can be captured at any sampling frequency (i.e., on a continuous scale), we chose to focus on the most representative values reported in previous literature. In addition, collecting human gait below 50 Hz would certainly alter not only DFA results but also mean and CV, and collecting above 240 Hz would increase processing time. Finally, there is little reason to think that DFA results from data sampled at 120 Hz would significantly differ from data sampled at a slightly lower frequency (e.g., 100 Hz), because our results at 240 Hz or 120 Hz were very similar. As mentioned earlier, walking speed and the coefficient of variation of time series may also play a role. Future studies should investigate the reliability of DFA results at different walking velocity. Another limitation is that we only considered treadmill walking, but our conclusions may not hold true for overground walking. Note that the study of fractal dynamics during overground walking is often performed on data captured with small accelerometers or footswitches [1,4,7,10,18–20]. Footswitches in particular–while limited in capturing only temporal variables such as stride time intervals–are often capable of higher sampling frequency (e.g., data is often collected at 1000 Hz or more). A final limitation of this study was that we focused solely on the scaling exponent α-DFA, and did not test other techniques. While this may be considered a limitation, our goal in this paper was to provide guidelines specifically related to the application of DFA to spatiotemporal variables. Previous studies have already compared the effect of sampling frequency on other measures of gait [38], and future studies may use our data (S1 Dataset) to ask other questions related to the reliability of gait parameters during treadmill walking.

In conclusion, sampling frequency seems to have little effect on α-DFA applied to spatiotemporal variables during treadmill walking. Overall, stride intervals seem to provide more reliable results than step intervals. While no significant differences were observed between conditions, a small trend toward lower α-DFA values with lower sampling frequencies lead us to recommend that data should be collected at around 120 Hz. This seems to be the best compromise between precise event detection and reduced processing time.

## Supporting information

**S1 Dataset. Raw data.** Spatiotemporal time series from 14 healthy young adults walking on a treadmill at their self-selected speed, in different conditions characterized by different sampling frequency of the 3D motion capture system. Conditions '240ds120' and '240ds60' correspond to the downsampled data from 240 Hz to 120 Hz and 60 Hz, respectively.
(MAT)

## Acknowledgments

The authors thank the volunteers for their participation, Daniel Jaravata for his help with data collection, Ben Senderling and William Denton for technical support, and the reviewers for their valuable comments.

## Author Contributions

**Conceptualization:** Vivien Marmelat.

**Data curation:** Austin Duncan, Shane Meltz.

**Formal analysis:** Vivien Marmelat, Shane Meltz.

**Investigation:** Austin Duncan, Shane Meltz.

**Project administration:** Vivien Marmelat.

**Resources:** Vivien Marmelat.

**Supervision:** Vivien Marmelat.

**Validation:** Vivien Marmelat.

**Visualization:** Vivien Marmelat, Shane Meltz.

**Writing – original draft:** Vivien Marmelat.

**Writing – review & editing:** Vivien Marmelat, Austin Duncan, Shane Meltz.

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
