## [Decision Letter · Decision Letter 0]

7 Aug 2019

PONE-D-19-16440

Effect of sampling frequency on fractal fluctuations during treadmill walking

PLOS ONE

Dear Dr. Marmelat,

Thank you for submitting your manuscript to PLOS ONE. After careful consideration, we feel that it has merit but does not fully meet PLOS ONE’s publication criteria as it currently stands. Therefore, we invite you to submit a revised version of the manuscript that addresses the points raised during the review process.

The reviewers and I feel that a revised version of this manuscript will add to the existing literature, please address the comments below and we will consider your revised manuscript at that time. 

We would appreciate receiving your revised manuscript by Sep 21 2019 11:59PM. To enhance the reproducibility of your results, we recommend that if applicable you deposit your laboratory protocols in protocols.io, where a protocol can be assigned its own identifier (DOI) such that it can be cited independently in the future. For instructions see: http://journals.plos.org/plosone/s/submission-guidelines#loc-laboratory-protocols

We look forward to receiving your revised manuscript.

Kind regards,

Eric R. Anson

Academic Editor

PLOS ONE

Reviewers' comments:

Reviewer's Responses to Questions

**Comments to the Author**

1. Is the manuscript technically sound, and do the data support the conclusions?

Reviewer #1: Partly

Reviewer #2: Yes

2. Has the statistical analysis been performed appropriately and rigorously? 

Reviewer #1: No

Reviewer #2: Yes

3. Have the authors made all data underlying the findings in their manuscript fully available?

Reviewer #1: Yes

Reviewer #2: Yes

4. Is the manuscript presented in an intelligible fashion and written in standard English?

Reviewer #1: Yes

Reviewer #2: Yes

5. Review Comments to the Author

Reviewer #1: The authors examine whether capturing spatiotemoral gait data on a treadmill at different sampling frequencies affects either the (1) magnitude or (2) reliability of fractal indices (DFA alpha) from various time series (step & stride length, step & stride time, step & stride speed). They report data from 14 participants who walked on a treadmill at their preferred speed during each of three 15-minute bouts. Each bout was captured at a different sampling frequency (60 Hz, 120 Hz, 240 Hz) in a randomized order. The study’s purpose was well-justified and the paper was well-written. The protocol and data processing methods were sufficiently described. Nevertheless, I have two primary concerns with the study’s methods: (1) sample size and (2) data analysis approach. The following specific comments are not intended to be critical, but rather are intended to help the authors improve the study and/or paper.

(Methods section, line 86 & line 144) The authors did not provide justification for their sample size and I believe the study was underpowered to accurately estimate measurement reliability. Per line 86, 17 individuals participated but per line 144, data from 3 participants were excluded, which further diminishes the sample size. The effective sample size was n = 14. Given the authors’ use of the following guidelines for interpreting ICCs (ICC < .50 = poor reliability, ICC of .50 to .75 = moderate reliability, ICC of .75 to .90 = good reliability, ICC > .90 = excellent reliability), I might have expected the authors to power the study in a way that the sample size would have been sufficient to detect reliability coefficients of .75 or higher against a minimally acceptable reliability coefficient of .50. Using those parameters across 3 measurement conditions, approximately 27 or more participants would have been required to adequately power the study (Walter SD, Eliasziw M, Donner A. Sample size and optimal designs for reliability studies. Statistics in Medicine. 1998;17:101-110.).

(Methods section, lines 133-135) In my opinion, the study design is more representative of an alternate forms reliability study, i.e., evaluating agreement between methods of measurement. Bland-Altman plots and limits of agreement analyses are perhaps better designed to address questions of agreement than are ICCs (Bland JM, Altman DG. A note on the use of the intraclass correlation coefficient in the evaluation of agreement between two methods of measurement. Computers in Biology and Medicine. 1990;20(5):337-340). Would the authors consider re-analyzing their data with Bland-Altman plots either instead of or as a supplemental analysis to the ICCs?

(Results section, lines 153-155) Per my previous comment regarding sample size, I would suggest the study was underpowered to detect accurate reliability coefficients and is a major reason the 95% confidence intervals reported in Table 1 are so wide. Informing the reader that measuring DFA alpha has “poor to moderate” or “poor to good” reliability may therefore be misleading. Those interpretations would be much more valid if the study was appropriately powered. Additionally, analyzing limits of agreement and/or systematic bias across measurement methods with Bland-Altman analyses may provide an alternative interpretation.

(Results section, Table 1) The step speed and stride speed DFA alpha values are approximately .50 (i.e., white noise) with very small standard deviations, suggesting little between-subject variability. ICC values are artificially deflated when between-subject variability is low, which would partially explain why those ICCs are so poor. I suspect that if the authors create Bland-Altman plots and analyze limits of agreement between the 60 Hz, 120 Hz and 240 Hz methods of measurement, the results will show strong agreement between methods with little to no evidence of systematic bias.

(Discussion section, line 184) Per my preceding comment, it is potentially misleading to conclude that DFA alpha “values were not reliable between conditions using different sampling frequencies.” If results are similar with an adequately powered sample size and/or systematic biases in the measures are apparent with Bland-Altman analyses, then the conclusion would be more valid.

(Discussion section, lines 205-209) The discussion content represented in lines 205-209 is potentially misleading for reasons listed above. The discussion content will be acceptable—in my opinion—only if the ICCs remain low with an adequately powered sample and if Bland-Altman analyses provide evidence of poor agreement (systematic bias) between methods.

(Discussion section, lines 225-249) The authors neglected the major limitation in their study: sample size. If the authors choose not to collect data from more participants and revise the manuscript accordingly, then the sample size limitation must be addressed as a major limitation to the study’s findings.

Reviewer #2: General: This paper provides an important technical contribution to the field of gait measurement. The authors examined the role of sample frequency and downsampling on prominently used gait metrics. Their results highlight the strengths and challenges of different sampling/processing techniques that people working in this area should be aware of. My comments below are minor and are aimed to help provide clarification in a few places in the text.

Comments and suggestions to the authors:

ABSTRACT

Lines 29-30: “Our results suggest that sampling frequency (between 60 and 240 Hz) does not significantly alter DFA”. Does the data support this? Lines 27-28 said “Intraclass correlation analysis revealed a poor reliability of DFA results between conditions using different sampling frequencies.” This seems to be a disconnect and could use some clarification in the text. (See my second comment in the Discussion section below for more context on this)

Line 32: The word “optimal” implies a data driven approach that provides support for such a claim. Since it appears that the relationship between detection accuracy and processing time was not quantified in this study, I recommend removing the word “optimal” from this sentence.

INTRODUCTION

Lines 39-44: You accurately identify that stride-to-stride fluctuations can become anti-persistent in various conditions. However, it would provide a more holistic view of work in this area if you were to also include text and references indicating that stride-to-stride fluctuations can also become more persistent in some conditions. This inclusion is important because it demonstrates gait fluctuations are modifiable in either direction (toward anti-persistence or persistence), which could be valuable for readers of this paper. In addition to some of the references you already cited, below are some additional references to consider including that show a shift toward persistence.

Rhea, C. K., Kiefer, A. W., Wittstein, M. W., Leonard, K. B., MacPherson, R. P., Wright, W. G., & Haran, F. J. (2014). Fractal gait patterns are retained after entrainment to a fractal stimulus. PLOS ONE, 9(9), e106755.

Rhea, C. K., Kiefer, A. W., D’Andrea, S. E., Warren, W. H., & Aaron, R. K. (2014). Entrainment to a real time fractal visual stimulus modulates fractal gait dynamics. Human Movement Science, 36, 20-34.

Wittstein, M. W., Starobin, J. M., Schmitz, R. J., Shulz, S. J., Haran, F. J., & Rhea, C. K. (2019). Cardiac and gait rhythms in healthy younger and older adults during treadmill walking tasks. Aging Clinical and Experimental Research, 31(3), 367-375.

Line 67-69: Please add a reference to support this statement.

METHODS

Lines 87-88: I assume this was verified via self-report, not medical record examination, correct? If so, please change this sentence to read “All participants self-reported no cognitive, neurological, muscular, or orthopedic impairments.”

RESULTS

Line 144: Please report the age (mean+/-SD) and gender of the 14 participants who were included in the analysis since these data reported in the Methods section pertains to the original 17 included in the study.

Line 166: Please add the degrees of freedom prior to your F-statistic.

DISCUSSION

Lines 182-184: It’s not clear how the following can co-exist for α-DFA:

“i) in general, mean, CV and α-DFA values of all spatiotemporal variables were similar between conditions, whether the data was collected at different sampling frequencies or downsampled, ii) α-DFA values were not reliable between conditions using different sampling frequencies”

How can α-DFA be similar between conditions collected at different sampling frequencies AND not reliable between conditions using different sampling frequencies? Some clarification needs to be made for these two points. You bring up this point again in lines 205-206. Your position on this appears to be accurate, but it would be helpful to readers if you can describe how data can be similar between conditions, yet unreliable. Many people many not understand how those two things can independently fluctuate.

Lines 192-193: Please describe how your data supports these two statements:

“Our results suggest that when the research question focuses on within-group or between-group comparisons, a sampling frequency as low as 60 Hz may be able to capture differences. While the reductions in α-DFA were not significant, 120 Hz may allow for more precise event detection.”

Line 233: The use of “dramatically” is relative and likely not a problem for most computers given current processing speeds. It will, however, increase processing speeds. Thus, I recommend removing “dramatically” from this sentence.

6. PLOS authors have the option to publish the peer review history of their article (what does this mean?). If published, this will include your full peer review and any attached files.

Reviewer #1: No

Reviewer #2: Yes: Christopher K. Rhea

---

## [Author Response · Author response to Decision Letter 0]

7 Oct 2019

Response to reviewers

We thank the editor and reviewers for their comments. We have addressed each comment and made revisions that we hope will be satisfactory. 

Editor’s comment

We have revised the manuscript format according to PLOS ONE’s style requirements, including files names. 

We apologize for missing this in the first place. We have revised accordingly the file title, added the title of the subsection ‘Supporting information’ and revised in-text citations. 

Reviewer’s comment

Reviewer #1: The authors examine whether capturing spatiotemporal gait data on a treadmill at different sampling frequencies affects either the (1) magnitude or (2) reliability of fractal indices (DFA alpha) from various time series (step & stride length, step & stride time, step & stride speed). They report data from 14 participants who walked on a treadmill at their preferred speed during each of three 15-minute bouts. Each bout was captured at a different sampling frequency (60 Hz, 120 Hz, 240 Hz) in a randomized order. The study’s purpose was well-justified and the paper was well-written. The protocol and data processing methods were sufficiently described. Nevertheless, I have two primary concerns with the study’s methods: (1) sample size and (2) data analysis approach. The following specific comments are not intended to be critical, but rather are intended to help the authors improve the study and/or paper.

Thank you for your insightful comments. We have made our best to address each of them. 

(Methods section, line 86 & line 144) The authors did not provide justification for their sample size and I believe the study was underpowered to accurately estimate measurement reliability. Per line 86, 17 individuals participated but per line 144, data from 3 participants were excluded, which further diminishes the sample size. The effective sample size was n = 14. Given the authors’ use of the following guidelines for interpreting ICCs (ICC < .50 = poor reliability, ICC of .50 to .75 = moderate reliability, ICC of .75 to .90 = good reliability, ICC > .90 = excellent reliability), I might have expected the authors to power the study in a way that the sample size would have been sufficient to detect reliability coefficients of .75 or higher against a minimally acceptable reliability coefficient of .50. Using those parameters across 3 measurement conditions, approximately 27 or more participants would have been required to adequately power the study (Walter SD, Eliasziw M, Donner A. Sample size and optimal designs for reliability studies. Statistics in Medicine. 1998;17:101-110.).

We agree that this study may be underpowered, and we have now explicitly stated this as a limitation in the Discussion. 

According to Table II in Walter et al. (1998), for n=3, ρ0 = 0.5 (i.e., testing the null hypothesis ρ = ρ0, where ρ0 is the minimally acceptable level of reliability), and ρ1 = 0.8 (where ρ1 is a specific underlying value of ρ under H1, i.e., ρ > ρ0), only 15 subjects would be enough. We agree that, based on our guidelines for interpreting ICC values (i.e., ρ1 = 0.75), a sample size of 25 subjects should be more adequate. We also want to stress that our guidelines are very conservatives in estimating reliability, because we considered the lowest 95% confidence interval for our comparisons and conclusions. This approach may be more pragmatic, in that we reduce the risk of estimating a measure as ‘reliable’ while it may have very large CIs. 

(Methods section, lines 133-135) In my opinion, the study design is more representative of an alternate forms reliability study, i.e., evaluating agreement between methods of measurement. Bland-Altman plots and limits of agreement analyses are perhaps better designed to address questions of agreement than are ICCs (Bland JM, Altman DG. A note on the use of the intraclass correlation coefficient in the evaluation of agreement between two methods of measurement. Computers in Biology and Medicine. 1990;20(5):337-340). Would the authors consider re-analyzing their data with Bland-Altman plots either instead of or as a supplemental analysis to the ICCs?

We thank the reviewer for this comment. We agree that Bland-Altman plots and limits of agreement are relevant complementary analyses to our work. We have conducted this analysis on DFA values, for all possible pairs of conditions, for all spatiotemporal variables (left and right side). As for the other analyses, there was no differences between left and right side, so we reported results for right side only for the sake of clarity. While Bland-Altman plots are a clear visual representation of agreement between methods, due to the number of comparisons in this study we decided to report the bias within a table (Table 2 in the revised manuscript), defined as the mean difference between paired measures, and the 95% limits of agreement, defined as the bias � 1.96 standard deviation of the differences. 

As recommended by Giavarina (2015), we defined a priori the limits of agreement (LoA) expected (limits of maximum acceptable differences) to 0.05, i.e., for a bias of 0 the α-DFA values measured in one condition are deemed similar to those measured in another condition if they fall within a range of 0.1. 

(Results section, lines 153-155) Per my previous comment regarding sample size, I would suggest the study was underpowered to detect accurate reliability coefficients and is a major reason the 95% confidence intervals reported in Table 1 are so wide. Informing the reader that measuring DFA alpha has “poor to moderate” or “poor to good” reliability may therefore be misleading. Those interpretations would be much more valid if the study was appropriately powered. Additionally, analyzing limits of agreement and/or systematic bias across measurement methods with Bland-Altman analyses may provide an alternative interpretation.

Bland-Altman analyses did not show any systematic bias, although the bias was higher when comparing condition 240 vs. ds60 (compared to condition 240 vs. ds120), for all spatiotemporal variables. 

In addition, the limits of agreement were all above the (a priori determined) threshold of 0.05, expect when comparing condition 240 to ds120. This result may actually strengthen our conclusion that using a sampling frequency of 240 Hz does not provide more benefits than 120 Hz, but that 60 Hz alters DFA results. 

(Results section, Table 1) The step speed and stride speed DFA alpha values are approximately .50 (i.e., white noise) with very small standard deviations, suggesting little between-subject variability. ICC values are artificially deflated when between-subject variability is low, which would partially explain why those ICCs are so poor. I suspect that if the authors create Bland-Altman plots and analyze limits of agreement between the 60 Hz, 120 Hz and 240 Hz methods of measurement, the results will show strong agreement between methods with little to no evidence of systematic bias.

Results from the Bland-Altman analysis supported the results from ICCs and our original conclusions: while the bias was very low for both step speed and stride speed when comparing condition 240 to both conditions 120 and 60, the 95% LoA were still very high (between 0.16 and 0.18), well above the threshold of 0.05. In our opinion, the low ICC results from the fact that despite relatively low between-subject variability (at least lower than for stride and step time and length), there was no consistent trend in DFA changes (cf. figure 2). 

(Discussion section, line 184) Per my preceding comment, it is potentially misleading to conclude that DFA alpha “values were not reliable between conditions using different sampling frequencies.” If results are similar with an adequately powered sample size and/or systematic biases in the measures are apparent with Bland-Altman analyses, then the conclusion would be more valid.

While the limitations about sample sizes remains, we think the results from the Bland-Altman analysis support our original conclusion. In particular, the Bland-Altman analysis allowed us to study pairwise comparisons of conditions. Interestingly, the data from condition ds60 compared to ds120 showed 95% LoA above the (a priori) threshold of 0.05, with a negative bias, suggesting that ds60 produced lower DFA values and reinforcing our impression that sampling at 60 Hz is not recommended. 

 (Discussion section, lines 205-209) The discussion content represented in lines 205-209 is potentially misleading for reasons listed above. The discussion content will be acceptable—in my opinion—only if the ICCs remain low with an adequately powered sample and if Bland-Altman analyses provide evidence of poor agreement (systematic bias) between methods.

While we were not able to increase the sample size, the results from the Bland-Altman analysis seems to support the conclusion that there are few differences between data sampled at 240 Hz and downsampled at 120 Hz, but that 60 Hz is less ‘similar’ than both 240 and 120. In particular, there was a systematic bias for DFA values from ds60, which were always lower than ds120 (expect for Stride speed). 

(Discussion section, lines 225-249) The authors neglected the major limitation in their study: sample size. If the authors choose not to collect data from more participants and revise the manuscript accordingly, then the sample size limitation must be addressed as a major limitation to the study’s findings.

We agree that this study may be underpowered. We have now explicitly stated this as a limitation in the Discussion. 

Reviewer #2: General: This paper provides an important technical contribution to the field of gait measurement. The authors examined the role of sample frequency and downsampling on prominently used gait metrics. Their results highlight the strengths and challenges of different sampling/processing techniques that people working in this area should be aware of. My comments below are minor and are aimed to help provide clarification in a few places in the text.

Comments and suggestions to the authors:

ABSTRACT

Lines 29-30: “Our results suggest that sampling frequency (between 60 and 240 Hz) does not significantly alter DFA”. Does the data support this? Lines 27-28 said “Intraclass correlation analysis revealed a poor reliability of DFA results between conditions using different sampling frequencies.” This seems to be a disconnect and could use some clarification in the text. (See my second comment in the Discussion section below for more context on this)

The first sentence referred to ANOVA results while the second referred to ICC results. In other words, while there was no statistical significant differences between conditions from the ANOVAs, the individual values were not ‘similar’ between conditions. 

Line 32: The word “optimal” implies a data driven approach that provides support for such a claim. Since it appears that the relationship between detection accuracy and processing time was not quantified in this study, I recommend removing the word “optimal” from this sentence.

We agree with this comment and the sentence has been revised. 

INTRODUCTION

Lines 39-44: You accurately identify that stride-to-stride fluctuations can become anti-persistent in various conditions. However, it would provide a more holistic view of work in this area if you were to also include text and references indicating that stride-to-stride fluctuations can also become more persistent in some conditions. This inclusion is important because it demonstrates gait fluctuations are modifiable in either direction (toward anti-persistence or persistence), which could be valuable for readers of this paper. In addition to some of the references you already cited, below are some additional references to consider including that show a shift toward persistence.

Rhea, C. K., Kiefer, A. W., Wittstein, M. W., Leonard, K. B., MacPherson, R. P., Wright, W. G., & Haran, F. J. (2014). Fractal gait patterns are retained after entrainment to a fractal stimulus. PLOS ONE, 9(9), e106755.

Rhea, C. K., Kiefer, A. W., D’Andrea, S. E., Warren, W. H., & Aaron, R. K. (2014). Entrainment to a real time fractal visual stimulus modulates fractal gait dynamics. Human Movement Science, 36, 20-34.

Wittstein, M. W., Starobin, J. M., Schmitz, R. J., Shulz, S. J., Haran, F. J., & Rhea, C. K. (2019). Cardiac and gait rhythms in healthy younger and older adults during treadmill walking tasks. Aging Clinical and Experimental Research, 31(3), 367-375.

That is a good point, we have revised the introduction to reflect that change and included additional references, including those suggest by the reviewer. 

Line 67-69: Please add a reference to support this statement.

The following reference has been added:

Wijnants ML, Cox RFA, Hasselman F, Bosman AMT, Van Orden G. Does sample rate introduce an artifact in spectral analysis of continuous processes? Front Physio. 2013; 3:495.doi: 10.3389/fphys.2012.00495.

METHODS

Lines 87-88: I assume this was verified via self-report, not medical record examination, correct? If so, please change this sentence to read “All participants self-reported no cognitive, neurological, muscular, or orthopedic impairments.”

This sentence has been revised accordingly. 

RESULTS

Line 144: Please report the age (mean+/-SD) and gender of the 14 participants who were included in the analysis since these data reported in the Methods section pertains to the original 17 included in the study.

We have included the following sentence:

Data from the remaining 14 participants (Age 23.9 ± 2.8 years, 5 females) were further processed.

Line 166: Please add the degrees of freedom prior to your F-statistic.

Thank you for noticing this typo, it has been revised accordingly. 

DISCUSSION

Lines 182-184: It’s not clear how the following can co-exist for α-DFA:

“i) in general, mean, CV and α-DFA values of all spatiotemporal variables were similar between conditions, whether the data was collected at different sampling frequencies or downsampled, ii) α-DFA values were not reliable between conditions using different sampling frequencies”

How can α-DFA be similar between conditions collected at different sampling frequencies AND not reliable between conditions using different sampling frequencies? Some clarification needs to be made for these two points. You bring up this point again in lines 205-206. Your position on this appears to be accurate, but it would be helpful to readers if you can describe how data can be similar between conditions, yet unreliable. Many people many not understand how those two things can independently fluctuate.

Indeed, we may have failed to give a simple description of our main conclusion, which is that DFA values may not be reliable from one condition to the other, despite the absence of significant differences between conditions. We have revised this section of our discussion by including to which statistical analyses each conclusion is based on, to hopefully better explain how we interpret our results. 

Lines 192-193: Please describe how your data supports these two statements:

“Our results suggest that when the research question focuses on within-group or between-group comparisons, a sampling frequency as low as 60 Hz may be able to capture differences. While the reductions in α-DFA were not significant, 120 Hz may allow for more precise event detection.”

The first sentence is supported by results from the ANOVA: there were no statistically significant differences between conditions 240 and 60. We attempted to explain why 120 Hz would allow more precise estimation of event detection in the following sentences describing how walking speed and CV also influence the choice of sampling frequency. 

Line 233: The use of “dramatically” is relative and likely not a problem for most computers given current processing speeds. It will, however, increase processing speeds. Thus, I recommend removing “dramatically” from this sentence.

We agree and have revised accordingly.

---

## [Decision Letter · Decision Letter 1]

23 Oct 2019

PONE-D-19-16440R1

Effect of sampling frequency on fractal fluctuations during treadmill walking

PLOS ONE

Dear Dr. Marmelat,

Thank you for submitting your manuscript to PLOS ONE. After careful consideration, we feel that it has merit but does not fully meet PLOS ONE’s publication criteria as it currently stands. Therefore, we invite you to submit a revised version of the manuscript that addresses the points raised during the review process.

The revised version of this manuscript has been reviewed and both reviewers felt that the authors satisfactorily addressed their concerns.  There remain 3 minor details that should be addressed.

1) It seems from context that the statement on line 264-265 is specific to DS60, but this is not clear in the statement.  Please clarify.

2) Table 1: Leading zero missing for ICC 95% CI for Step Length  [0.811-969] should probably be [0.811-0.969]

3) Table 2: Leading zero missing for a-DFA Stride Time in Table 2 (.14) should be (0.14)

We would appreciate receiving your revised manuscript by Dec 07 2019 11:59PM. To enhance the reproducibility of your results, we recommend that if applicable you deposit your laboratory protocols in protocols.io, where a protocol can be assigned its own identifier (DOI) such that it can be cited independently in the future. For instructions see: http://journals.plos.org/plosone/s/submission-guidelines#loc-laboratory-protocols

We look forward to receiving your revised manuscript.

Kind regards,

Eric R. Anson

Academic Editor

PLOS ONE

Reviewers' comments:

Reviewer's Responses to Questions

**Comments to the Author**

1. If the authors have adequately addressed your comments raised in a previous round of review and you feel that this manuscript is now acceptable for publication, you may indicate that here to bypass the “Comments to the Author” section, enter your conflict of interest statement in the “Confidential to Editor” section, and submit your "Accept" recommendation.

Reviewer #1: All comments have been addressed

Reviewer #2: All comments have been addressed

2. Is the manuscript technically sound, and do the data support the conclusions?

Reviewer #1: Yes

Reviewer #2: Yes

3. Has the statistical analysis been performed appropriately and rigorously? 

Reviewer #1: Yes

Reviewer #2: Yes

4. Have the authors made all data underlying the findings in their manuscript fully available?

Reviewer #1: Yes

Reviewer #2: Yes

5. Is the manuscript presented in an intelligible fashion and written in standard English?

Reviewer #1: Yes

Reviewer #2: Yes

6. Review Comments to the Author

Reviewer #1: I appreciate the thoroughness of the authors' response to reviewers and appreciate their revisions to the manuscript. The paper will provide a valuable technical contribution to gait researchers.

Reviewer #2: The authors did an excellent job revising this manuscript. They are commended on their effort and the manuscript is recommended for publication in its current form.

7. PLOS authors have the option to publish the peer review history of their article (what does this mean?). If published, this will include your full peer review and any attached files.

Reviewer #1: No

Reviewer #2: No

---

## [Author Response · Author response to Decision Letter 1]

23 Oct 2019

Response to reviewers

We thank the editor and reviewers for their comments. We have addressed each comment and revised the small typos. 

1) It seems from context that the statement on line 264-265 is specific to DS60, but this is not clear in the statement. Please clarify.

We have clarified the statement as follow:

This suggests that sampling motion capture at 60 Hz (i.e., as in condition DS60) may lead to less accurate α-DFA values, assuming condition 240 as the gold-standard.

2) Table 1: Leading zero missing for ICC 95% CI for Step Length [0.811-969] should probably be [0.811-0.969]

Revised. 

3) Table 2: Leading zero missing for a-DFA Stride Time in Table 2 (.14) should be (0.14)

Revised.

---

## [Editor Report · Decision Letter 2]

28 Oct 2019

Effect of sampling frequency on fractal fluctuations during treadmill walking

PONE-D-19-16440R2

Dear Dr. Marmelat,

We are pleased to inform you that your manuscript has been judged scientifically suitable for publication and will be formally accepted for publication once it complies with all outstanding technical requirements.

With kind regards,

Eric R. Anson

Academic Editor

PLOS ONE
---

## [Editor Report · Acceptance letter]

30 Oct 2019

PONE-D-19-16440R2 

Effect of sampling frequency on fractal fluctuations during treadmill walking 

Dear Dr. Marmelat:

I am pleased to inform you that your manuscript has been deemed suitable for publication in PLOS ONE. Congratulations! Your manuscript is now with our production department. 

With kind regards,

on behalf of

Dr. Eric R. Anson 

Academic Editor

PLOS ONE